# CRISPR-Cas-Guided Mutagenesis of Chromosome and Virulence Plasmid in *Shigella flexneri* by Cytosine Base Editing

Atin Sharma,[a,b,c] Ruqiya Omer Aden,[a,b,c] Andrea Puhar,[a,b,c] David A. Cisneros[a,b]

[a]Department of Molecular Biology, Umeå University, Umeå, Sweden
[b]Umeå Centre for Microbial Research, Umeå University, Umeå, Sweden
[c]The Laboratory for Molecular Infection Medicine Sweden (MIMS), Umeå University, Umeå, Sweden

Andrea Puhar and David A. Cisneros contributed equally to this work as senior authors.

**ABSTRACT** *Shigella* is a Gram-negative bacterium that invades the human gut epithelium. The resulting infection, shigellosis, is the deadliest bacterial diarrheal disease. Much of the information about the genes dictating the pathophysiology of *Shigella*, both on the chromosome and the virulence plasmid, was obtained by classical reverse genetics. However, technical limitations of the prevalent mutagenesis techniques restrict the generation of mutants in a single reaction to a small number, preventing large-scale targeted mutagenesis of *Shigella* and the subsequent assessment of phenotype. We adopted a CRISPR-Cas-dependent approach, where a nickase Cas9 and cytidine deaminase fusion is guided by single guide RNA (sgRNA) to introduce targeted C→T transitions, resulting in internal stop codons and premature termination of translation. In proof-of-principle experiments using an *mCherry* fluorescent reporter, we were able to generate loss-of-function mutants in both *Escherichia coli* and *Shigella flexneri* with up to 100% efficacy. Using a modified fluctuation assay, we determined that under optimized conditions, the frequency of untargeted mutations introduced by the Cas9-deaminase fusion was in the same range as spontaneous mutations, making our method a safe choice for bacterial mutagenesis. Furthermore, we programmed the method to mutate well-characterized chromosomal and plasmid-borne *Shigella flexneri* genes and found the mutant phenotype to be similar to those of the reported gene deletion mutants, with no apparent polar effects at the phenotype level. This method can be used in a 96-well-plate format to increase the throughput and generate an array of targeted loss-of-function mutants in a few days.

**IMPORTANCE** Loss-of-function mutagenesis is critical in understanding the physiological role of genes. Therefore, high-throughput techniques to generate such mutants are important for facilitating the assessment of gene function at a pace that matches systems biology approaches. However, to our knowledge, no such method was available for generating an array of single gene mutants in an important enteropathogen—*Shigella*. This pathogen causes high morbidity and mortality in children, and antibiotic-resistant strains are quickly emerging. Therefore, determination of the function of unknown *Shigella* genes is of the utmost importance to develop effective strategies to control infections. Our present work will bridge this gap by providing a rapid method for generating loss-of-function mutants. The highly effective and specific method has the potential to be programmed to generate multiple mutants in a single, massively parallel reaction. By virtue of plasmid compatibility, this method can be extended to other members of *Enterobacteriaceae*.

**KEYWORDS** *Enterobacteriaceae*, loss of function mutagenesis, polar effects, polar mutagenesis

S*higella* is a Gram-negative bacterium that belongs to the *Enterobacteriaceae* family. *Shigella flexneri* is the most prevalent species resulting in disease—shigellosis, or

Address correspondence to Andrea Puhar, andrea.puhar@umu.se, or David A. Cisneros, david.cisneros@umu.se.

The authors declare no conflict of interest.

bacillary dysentery (1). Shigellosis is mostly a self-limiting disease in healthy adults but poses a major threat to children, the elderly, and the immunocompromised (2). With an annual death toll of 212,000, of which 63,713 are children below the age of 5, shigellosis is the deadliest bacterial diarrheal disease (3). Very low infectious doses (10 to 100 bacteria), easy person-to-person spread, and the acute inflammatory response are responsible for such high rates of incidence and mortality, especially in low- and medium-income countries (4). Being an invasive enteric pathogen, *Shigella* gains access to the intestinal epithelial cells by attaching and inducing its own uptake. Once inside the cells, it lyses the vacuole, replicates, and by using actin-based motility, disseminates to neighboring cells (reviewed in reference 5). Virulence of *Shigella* requires a virulence plasmid-encoded type III secretion system (T3SS), which injects proteins into the host cell, aiding the bacterium in the establishment of infection through invasion, subversion of host defenses, and dissemination (reviewed in reference 6).

We recently resequenced the genome of *S. flexneri* serotype 5a strain M90T, a widely used laboratory reference strain, and mapped the global transcription start sites (7). Unsurprisingly, many transcription start sites led to the identification of genes that await functional characterization to reveal new and interesting information about *Shigella*'s physiology. Loss-of-function mutagenesis and subsequent assessment of phenotype is a common and perhaps the most convenient way to determine the function of unknown genes. The classic allele replacement technique, first described decades ago by Datsenko and Wanner (8), is still the method of choice for targeted mutations in *Shigella* (9–11), but it has some limitations. Since the success of this technique is directly dependent on the transformation of a linear DNA construct, the requirement of a large amount of DNA and electroporation limit its throughput. The two-step process, allele replacement and subsequent removal, leaves a scar sequence which, if not optimized for being in-frame, results in polar effects that affect phenotype assessment (12). Although PCR-mediated construction of a linear DNA construct is easy, it requires expensive long primers and electroporation, which adds to the final cost and acts as a technical bottleneck to increasing the throughput of mutagenesis.

CRISPR-Cas, a bacterial defense system against phages, has gained much attention for its use in genetic modifications (13). The *Streptococcus pyogenes* Cas9, the most commonly used protein for genome engineering, can be directed using guide RNA (gRNA) to introduce double-stranded DNA (dsDNA) breaks (14). In eukaryotes, this is followed by DNA repair, resulting in the generation of mutants, but in prokaryotes, a dsDNA break is lethal (15). Therefore, CRISPR-Cas based mutagenesis became the method of choice for targeted mutations in the eukaryotes, but its usage in prokaryotes was initially restricted to increasing the overall efficiency of allele replacement by positive selection (16, 17). However, later modifications resulted in the advent of base editors, or catalytically impaired Cas9 proteins fused to deaminases, resulting in base substitution and, eventually, loss-of-function mutations (18, 19). These methods avoid dsDNA breaks and result in markerless mutations. Moreover, the requirement of plasmid-borne single guide RNA (sgRNA) and effector proteins (Cas9 variants) eliminates the need of electroporation, thereby increasing the efficacy of the process, making it possible to multiplex, or generate massively parallel mutations (20–22).

Here, we describe a CRISPR-Cas-mediated base-editing method using an established cytidine base editor (20, 22) that can be programmed to generate loss-of-function mutations in *S. flexneri*. By mutating well-characterized *S. flexneri* genes of known phenotype/function, encoded on both the virulence plasmid and the chromosome, we show that the one-step method is easy, fast, and highly effective in inactivating genes. Utilizing this method, multiple genes in *S. flexneri* can be mutated in parallel, showing its great potential for high-throughput mutagenesis.

## RESULTS

**Construction of a two-plasmid-based system for expression of sgRNA and nCas9-AID.** As an effector protein, we selected a fusion of a catalytically inactive Cas9, nickase Cas9 (nCas9$^{D10A}$, bearing a mutation of D to A at position 10), and an ortholog of

activation-induced cytidine deaminase (AID), previously described as an effective base editor in the Gram-positive industrial bacterium *Corynebacterium glutamicum* (22). The fusion protein is guided to the target gene by single guide RNA (sgRNA; a 20-nucleotide target-specific "spacer" fused to a gRNA scaffold) where, within a window of $-20$ to $-16$ from the protospacer adjacent motif (PAM), the AID deaminates cytosine (C) to uracil (U) and the nCas9 nicks the unmodified strand, resulting in its replacement and a C→T (G→A on opposite strand) transition (Fig. 1A). This leads to conversion of internal CAG, CGA, and CAA codons on the coding strand or CCA on the noncoding strand into TAG, TGA, and TAA (stop codons), causing premature termination of translation and a loss-of-function mutation.

We generated pnCas9-AID, a medium-copy-number plasmid based on pSU19 (23), to express nCas9-AID (Fig. 1B). The SacB-encoding gene was cloned to facilitate curing of this plasmid (24). To express the sgRNA, we constructed a high-copy-number plasmid, pgRNA_AT, that uses a synthetic J23119 promoter. The *tac* promoter, used to drive the expression of nCas9-AID, is already known to function in *Shigella* (25). However, to determine J23119 activity in *Shigella*, superfolder green fluorescent protein (sfGFP) with a synthetic ribosome binding site was cloned in pgRNA_AT [pgRNA-(R)-sfGFP] and transformed into *S. flexneri* (Fig. S1 in the supplemental material). The expression of sfGFP, visualized by green fluorescence, showed constitutive activity of the J23119 promoter in *S. flexneri*.

To program a desired mutation, we used an online program, CRISPR-CBEI (26). It determines mutable sites and lists spacers required for the desired mutation in a target gene (Fig. 1C). We synthesized spacers as oligonucleotides with overhangs complementary to the ends of BsaI-digested pgRNA_AT to ensure directional cloning (Fig. 1D). The oligonucleotides were hybridized and phosphorylated prior to ligation and transformed into *E. coli* strain DH5$\alpha$. All the transformants obtained carried the required recombinant plasmid, as the cells transformed with the nonrecombinant plasmid were killed (22, 27).

**Base editing effectively inactivates chromosomal genes in *E. coli* and *S. flexneri*.** With optimal tools in hand, we tested our method by mutating a gene with an easy-to-read phenotype in *S. flexneri* and *E. coli* as a benchmark. We generated an *att*Tn7::*mCherry* insertion in *E. coli* MG1655 (*E. coli*::*mCherry*) and *S. flexneri* M90T 5a (*S. flexneri*::*mCherry*) using Tn7 transposition. We generated pgRNA_m2, pgRNA_m3, and pgRNA_m4 to introduce premature stop codons in mCherry at the 69th, 114th, and 98th codon positions. We also constructed pgRNA-X, expressing a nonbinding random spacer oligonucleotide.

We cotransformed *E. coli*::*mCherry* and *S. flexneri*::*mCherry* with pnCas9-AID or empty plasmid (pSU19) and with *mCherry*-targeting or control sgRNA plasmids (Fig. 2A). Cotransformants were selected as fluorescent colonies on selective medium containing glucose (Fig. S2). To optimize the time and conditions necessary for mutagenesis, we plated actively growing cultures of *E. coli*::*mCherry* and *S. flexneri*::*mCherry*, with or without IPTG induction. To determine the frequency of mutation (Fig. 2B and C), we counted the resulting colonies and scanned them for red fluorescence (Fig. 2D and E).

In both species, the basal frequency of mutagenesis (no nCas9-AID and control gRNA-X) was zero (Fig. 2B to E, condition 1), and an *mCherry* targeting sgRNA alone (condition 2) or nCas9-AID with sgRNA-X made no significant difference. This showed that gene-specific sgRNA alone or the nCas9-AID fusion bound to a control gRNA could not measurably introduce mutations. However, the frequency of mutagenesis in both species was significantly higher than in the control when nCas9-AID was expressed in the presence of two out of three *mCherry*-targeting sgRNAs (Fig. 2B to E, conditions 4 and 6). Although the frequencies were slightly lower in *S. flexneri* (median values of 85.7% and 66.7% compared to 100% in *E. coli*), they were in the same order of magnitude irrespective of the addition of IPTG (isopropyl $\beta$-D-thiogalactopyranoside). This result shows that base editing can be readily performed in *S. flexneri* with high efficiencies. However, a different effect was seen in the case of sgRNA_m3 (Fig. 2B and C, condition 5). While the median frequency in *E. coli* was lower (87.8%) but in the same order as with sgRNA_m2 and sgRNA_m4, the median frequency in *S. flexneri* (8.3%) was not

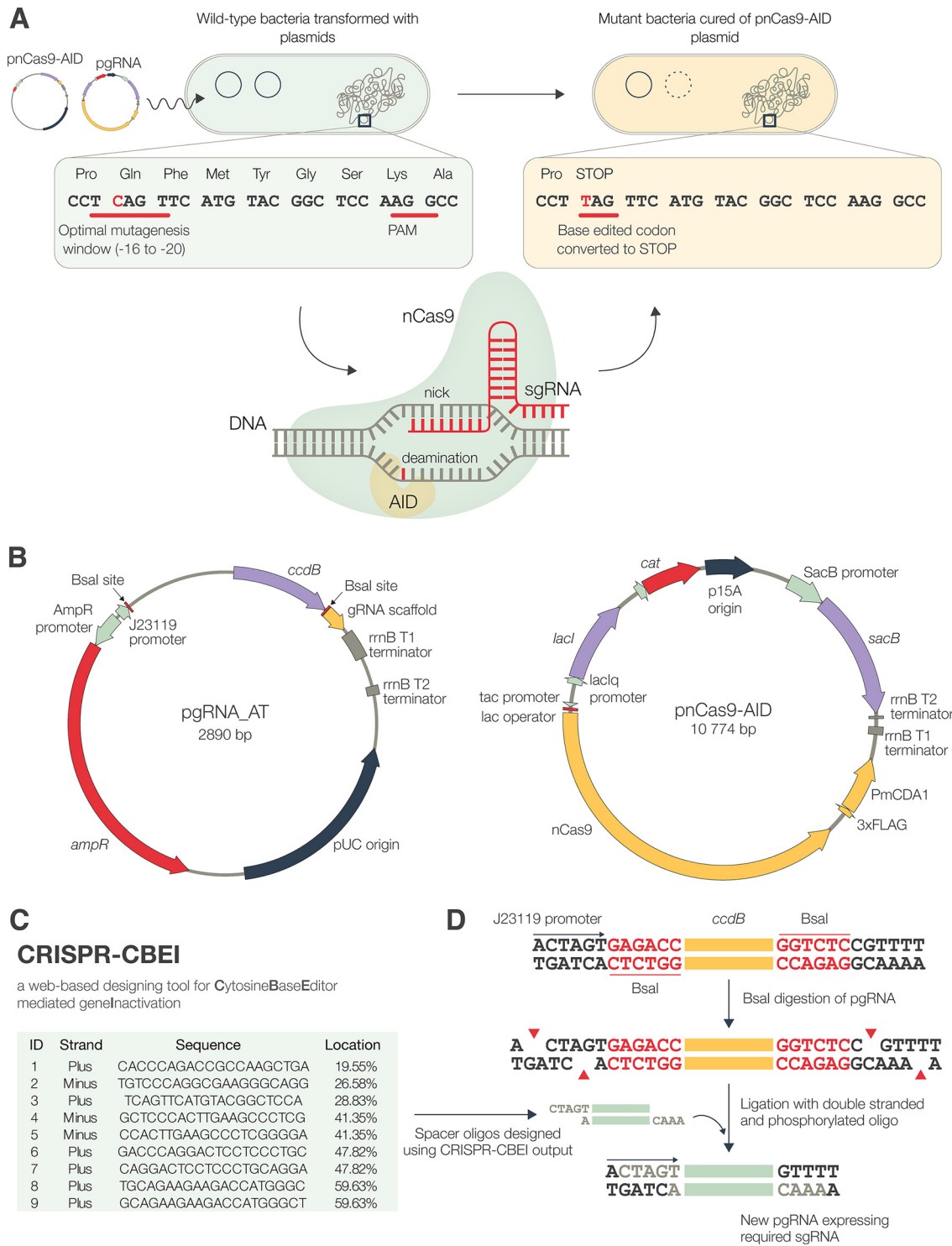

**FIG 1** Overview of the CRISPR-Cas-guided base-editing method. (A) Generation of mutants using *mCherry* as an example. After cotransformation with pnCas9-AID and pgRNA, the expression of Cas9-AID and a specific sgRNA introduces a premature stop codon in the target sequence. The PAM and the editable window are underlined, with the target C nucleotide shown in red. (B) Maps of plasmids pgRNA_AT and pnCas9-AID used for mutagenesis in *E. coli* and *S. flexneri*. (C and D) Output from CRISPR-CBEI for *mCherry* (C) and the general scheme for high-efficiency cloning of the spacers determined by CRISPR-CBEI (D).

significantly different from the control condition (Fig. 2C, condition 5). The addition of IPTG or longer incubation time did not improve the frequency (Fig. 2C and Fig. S3). Together, these results indicated that sgRNAs had significant differences in their ability to target *mCherry* that showed species-dependent variation. Effective mutagenesis,

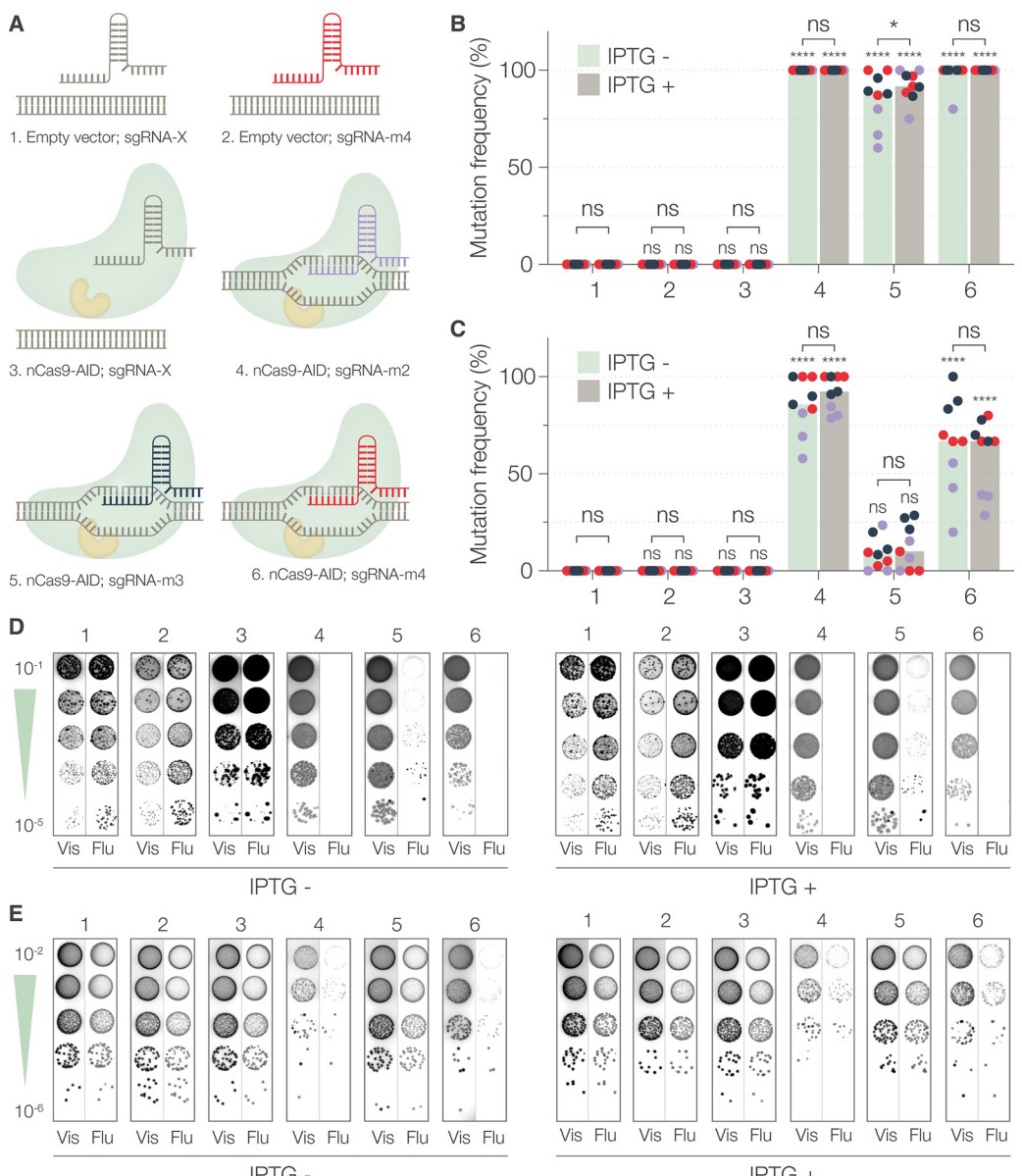

**FIG 2** Mutagenesis of chromosomally encoded *mCherry* in *E. coli* and *S. flexneri*. (A) Schematic representation of the six different conditions (plasmid combinations), numbered from 1 to 6, used in the experiment. (B and C) Mutation frequencies in *E. coli* (B) and *S. flexneri* (C), determined as the percent of the number of nonfluorescent (mutant) colonies to the total number of colonies, after 2 h of mutagenesis with or without IPTG induction. The numbers on the *x* axis correspond to the conditions represented in panel A. Each dot represents a technical replicate and is color coded to represent a biological replicate. The bars represent the median values from three independent experiments. Statistical significance was determined by performing two-way ANOVA followed by Sidák's multiple comparison test to determine the significance of each condition with respect to control conditions (condition 1) and to determine the significance of IPTG addition. ns, not significant; *, $P < 0.0332$; **, $P < 0.0021$; ***, $P < 0.0002$; ****, $P < 0.0001$. (D and E) Phenotypic determination of loss of fluorescence in *E. coli* (D) and *S. flexneri* (E). Representative visible (Vis) and fluorescence (Flu) images of colonies obtained by spotting logarithmic dilutions of induced (IPTG+) or uninduced (IPTG−) cultures after 2 h. The numbers 1 to 6 correspond to the conditions represented in panel A.

however, could still be performed in both species, surprisingly, without chemical induction of nCas9-AID expression.

We also determined the bacterial titer (CFU/mL) for all the time points tested under each condition (Fig. S3). The overall titer increased with time in both species, but there was no significant correlation across time points compared to the results for the control conditions (condition 1). Therefore, with the present data, we cannot strongly conclude whether carrying pgRNA derivatives, with or without pnCas9-AID, had specific

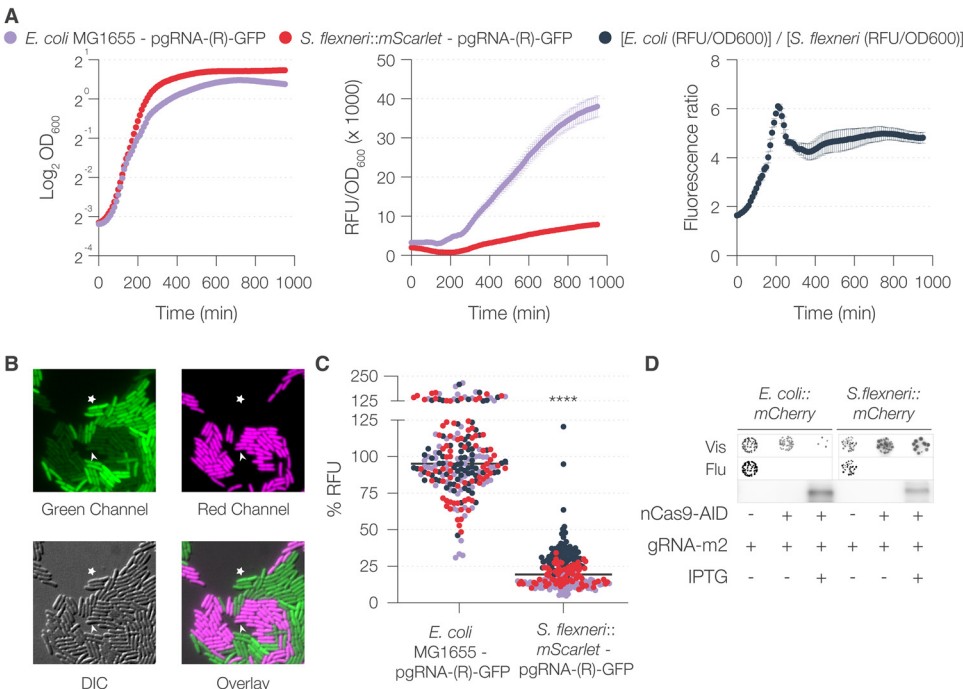

**FIG 3** Comparison of expression from pgRNA and pnCas9-AID in *E. coli* and *S. flexneri*. (A) Comparison of sfGFP fluorescence from pgRNA-(R)-GFP in *E. coli* and *S. flexneri::mScarlet*. Fluorescence and $OD_{600}$ values of bacterial cultures were measured simultaneously in a multimode plate reader, and the ratio (corrected fluorescence) was determined for each time point. Fold change in fluorescence was determined as the ratio of corrected fluorescence of *E. coli* and *S. flexneri::mScarlet* at each time point. RFU, relative fluorescence units. (B) Representative image of a 50:50 mixture of the cultures in the experiment whose results are shown in panel A taken with a fluorescence microscope. The arrowhead denotes a representative *S. flexneri* bacterium, and the asterisk marks a representative *E. coli* bacterium. (C) Quantification of fluorescence relative to the mean value of *E. coli* fluorescence. The horizontal line represents the median value. Each dot represents a technical replicate (fluorescent bacterium in one representative image) and is color coded to represent a biological replicate (experiments performed on different days). Statistical significance was determined by performing one-way ANOVA followed by Tukey's multiple comparison test. ns, not significant; *, $P < 0.0332$; **, $P < 0.0021$; ***, $P < 0.0002$; ****, $P < 0.0001$. (D) Expression of Cas9-AID fusion protein, determined by immunodetection using anti-Cas9 antibody. Mutagenesis of *mCherry*, guided by sgRNA_m2, was performed simultaneously. The outcome of the mutagenesis is shown as visible (Vis) and fluorescence (Flu) images of colonies obtained by spotting a $10^{-5}$ dilution of the cultures.

effects on the titers. Only a small, but measurable effect could be attributed to the addition of IPTG, which resulted in changes in growth rates or viability.

**nCas9-AID is highly effective at undetectable levels.** Differences in the efficacy of sgRNA_m3 in *E. coli* and *S. flexneri*, irrespective of IPTG induction, indicated that sgRNA, and not nCas9-AID expression, could be a factor responsible for the species-specific differences in the frequency of mutagenesis. To test this hypothesis, we transformed pgRNA-(R)-sfGFP into *E. coli* strain MG1655 and *S. flexneri::mScarlet* (*att*Tn7::*mScarlet* in *S. flexneri*) and measured the sfGFP fluorescence over time. The fluorescence was higher in *E. coli* (up to 6-fold) than in *S. flexneri::mScarlet* (Fig. 3A). This difference was confirmed at the single-cell level when the green fluorescence was quantified by microscopy (Fig. 3B and Fig. S4A). The expression of sfGFP from pgRNA-(R)-sfGFP was about four times higher in *E. coli* than in *S. flexneri* (Fig. 3C). This difference could be an outcome of several factors, including but not limited to plasmid copy number, promoter activity, and sgRNA stability. This difference in sfGFP expression in the context of pgRNA_AT could explain differences in targeting efficiencies between *E. coli* and *S. flexneri*.

We had already determined that induction of the expression of nCAs9-AID was, surprisingly, not necessary for efficient mutagenesis. We wanted to determine if different levels of nCas9-AID correlated with *mCherry* mutagenesis guided by sgRNA_m2. Our results showed that in both *E. coli* and *S. flexneri*, *mCherry* was mutated despite nCas9-AID levels being below the limit of detection as determined by Western blotting

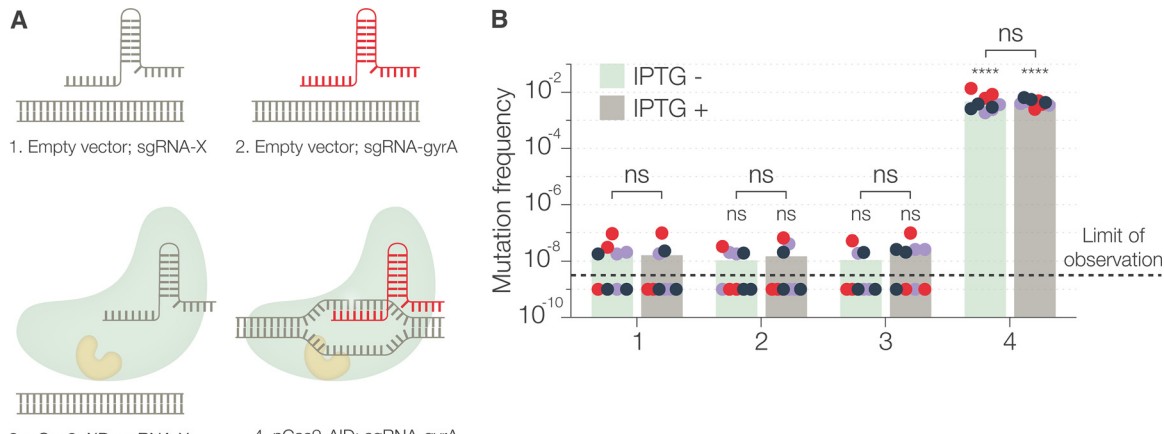

**FIG 4** Nalidixic acid resistance assay to compare the frequency of guided and random mutations. (A) Schematic representation of the four different conditions (plasmid combinations), numbered 1 to 4, used in the experiment. (B) Mutation frequency, determined as the ratio of the number of nalidixic acid-resistant (mutant) colonies to the total number of colonies, after 2 h of mutagenesis with and without IPTG induction. The numbers on the *x* axis correspond to the conditions represented in panel A. Each dot represents a technical replicate and is color coded to represent a biological replicate. In some technical replicates, the mutation frequency was below the limit of observation ($1 \times 10^{-9}$), as spotting high concentrations of the bacterial culture resulted in a bacterial mass instead of distinct antibiotic-resistant colonies. The bars represent the median values from three independent experiments. Statistical significance was determined by performing two-way ANOVA followed by Sidák's multiple comparison test for each condition with respect to the control (condition 1) and for IPTG addition. ns, not significant; *, $P < 0.0332$; **, $P < 0.0021$; ***, $P < 0.0002$; ****, $P < 0.0001$.

(Fig. 3D). nCas9-AID was detected upon IPTG induction, but it was accompanied by the appearance of bands below 100 kDa recognized by anti-Cas9 antibody, suggesting partial nCas9-AID degradation (Fig. S4B). Interestingly, a previous study used dCas9-AID fused to a degron to decrease its expression (21). We seem to have achieved the same outcome, albeit inadvertently. These results indicate that the nCas9-AID fusion protein is not very stable but highly effective, nonetheless, even at barely detectable levels.

**nCas9-AID-expressing strains are not hypermutators.** We suspected that leaky expression of nCas9-AID could alter the frequency of mutation at loci other than those targeted due to unspecific activity of the fused AID domain (28, 29). To test the basal mutation rate in *S. flexneri*, we performed the classic fluctuation assay using nalidixic acid (30). Point mutations in the gyrase-encoding gene, *gyrA*, render *Shigella* resistant to nalidixic acid (31). The frequency of appearance of such mutants can be used as a measure of the basal mutation rate. We designed an sgRNA (sgRNA-gyrA) that would cause a G→A transition at position 259, resulting in an Asp$_{87}$→Asn$_{87}$ substitution and resistance to nalidixic acid. *S. flexneri* strains carrying the backbone plasmid pSU19 instead of pnCas9-AID were used to distinguish between the mutation rates at the *gyrA* locus in the presence or absence of nCas9-AID (Fig. 4A). Mutagenesis was carried out as in the case of *mCherry*, but in addition to spotting on tryptic soy agar (TSA) plates, 10-fold serial dilutions of the cultures were also spotted on TSA plates containing nalidixic acid and the resulting colonies were counted.

The background rate of *gyrA* mutation in the absence of nCas9-AID, as a proxy measure of overall random mutation frequency, was in the order of $10^{-8}$ after 2 h of incubation (Fig. 4B). It was not significantly altered when IPTG was added. Importantly, the expression of nCas9-AID, in the presence of a nontargeting sgRNA (sgRNA-X), also did not significantly affect the mutation frequency. Even at later time points, the frequencies of mutation for strains expressing nCas9-AID with sgRNA-X were similar to the frequencies for those not expressing nCas9-AID at all (Fig. S4C). It was only in the presence of a *gyrA*-specific sgRNA (sgRNA-gyrA) that nCas9-AID was able to introduce nalidixic acid resistance mutations at a higher frequency. A nearly 5-log increase in the frequency of mutation confirmed the specificity of our method. The frequency of mutation, however, was not as high as with *mCherry* mutagenesis, highlighting again that

**TABLE 1** Details of characterized *S. flexneri* genes that were selected for mutagenesis[a]

| Gene | Location | Spacers | Function | Mutant phenotype | Reference(s) |
|------|----------|---------|----------|------------------|--------------|
| *vacJ* | Chromosome | Three spacers; same mutable site | Lysis of protrusion during intracellular dissemination | Dissemination defect | 36 |
| *icsA* | Virulence plasmid | Two spacers; different mutable sites | Actin-based motility | Dissemination defect | 35 |
| *icsB* | Virulence plasmid | Two spacers; different mutable sites | Postinvasion virulence factor | Dissemination defect (?) | 39, 40 |
| *mxiD* | Virulence plasmid | One spacer | Structural component of the injectisome | Invasion defect | 32 |

[a]The selected genes are found on the virulence plasmid or the chromosome. The variation in the number of spacers and their corresponding mutable sites is also mentioned.

efficacy can depend on target genes and sgRNAs. The low-level leaky expression and even induced expression of nCas9-AID were not mutagenic by themselves, probably due to the low stability of the fusion protein, as observed by Western blotting. This also suggested that the probability of introduction of unwanted unspecific mutations by nCas9-AID, under the conditions we tested, was in the same order of magnitude as for spontaneous mutations.

**Loss-of-function mutants generated by base editing have the same phenotype as gene deletion mutants.** Encouraged by our results, we decided to use our system to generate mutations in parallel in well-characterized virulence genes on both the chromosome and the virulence plasmid and verify the phenotypes of the mutants (Table 1 and Fig. 5A). We mutated *mxiD* in *S. flexneri* and performed a gentamicin protection assay to assess whether it had the same impact on invasion as reported earlier (32, 33). Wild-type and *S. flexneri mxiD(Q323X)* ($Q_{323}$ residue replaced with stop codon) cells were allowed to infect TC7 intestinal epithelial cells *in vitro*, and the extracellular bacteria were killed by gentamicin treatment. Bacterial invasion was quantified by lysing the TC7 cells to release intracellular bacteria that were protected from gentamicin and spotting logarithmic dilutions of the lysates on TSA plates (Fig. 5B). Wild-type (WT) bacteria successfully invaded TC7 cells, but we did not observe any invasion by *S. flexneri mxiD(Q323X)* (Fig. 5C). Since *mxiD* mutants fail to assemble a functional T3SS, they are unable to secrete the T3SS substrates (32). Indeed, we did not recover any secreted proteins in the supernatant of *S. flexneri mxiD(Q323X)* induced for secretion by Congo red (Fig. 5D) (34).

*icsA* and *vacJ* have been reported to be necessary for intracellular dissemination of *Shigella*, but they are not involved in invasion (35, 36). We first verified that base-edited mutants *S. flexneri icsA(Q59X)* and *S. flexneri vacJ(Q90X)* had no invasion defect (Fig. 5E). To assess intracellular dissemination, we used the plaque formation assay, where cells are infected with bacteria but the bacteria can only disseminate intracellularly due to the presence of gentamicin in the medium (37, 38). Successful intracellular dissemination results in the spread of bacteria and subsequent cell death, which appears as a zone of clearance in the cell monolayer (Fig. 5F). Successful invasion and subsequent dissemination were evident in case of WT *S. flexneri*, as numerous plaques of similar diameters were observed (Fig. 5G). However, no plaques were visible in the cases of *S. flexneri icsA(Q59X)* and *S. flexneri vacJ(Q90X)*, even at higher multiplicities of infection (MOIs), indicating an obvious dissemination defect.

Taken together, these results showed that the phenotypes of the mutants generated by our base-editing method were consistent with those reported in the literature.

**Base editing results in phenotypically nonpolar mutations.** Polar effects, or decreased expression of genes downstream from the site of mutation, are a common mutagenesis problem for polycistronic transcriptional units in bacteria (12). For example, polar effects of *icsB* mutation have been shown to impair intracellular dissemination in *Shigella* by affecting the transcription of downstream *ipa* genes (9, 39, 40). We generated two *icsB* mutants [*S. flexneri icsB(Q32X)* and *icsB(Q237X)*] with the premature stop codon introduced at two different sites (Fig. 5A). The *icsB* mutants had no apparent invasion defect (Fig. 5H) and there was no significant difference in the numbers and sizes of the plaques compared to those of WT bacteria (Fig. 5G and I). This indicated that the base-edited *icsB* mutant might be nonpolar.

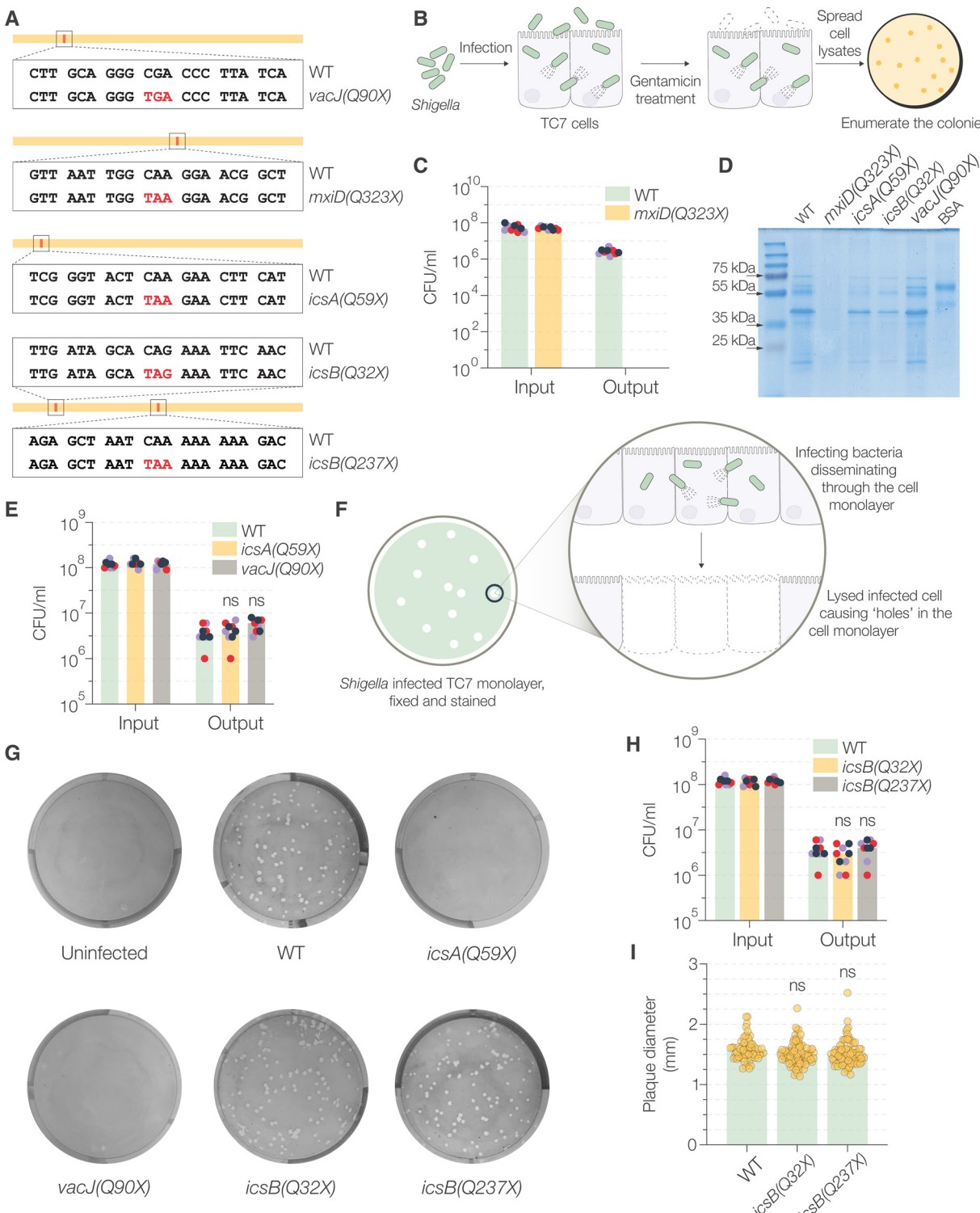

**FIG 5** CRISPR/Cas-guided mutagenesis of characterized *S. flexneri* genes and assessment of their phenotype. (A) Representation of the stop codons (highlighted in red) introduced by base editing. The total length of the gene is indicated by a line, and the red mark indicates the relative location of the mutable target site. (B) Schematic representation of the gentamicin protection assay used to assess the invasiveness of *Shigella*. (C) Invasiveness of *S. flexneri* WT and *mxiD(Q323X)*, determined by the number of bacterial CFU used to infect TC7 cells (input) and the number of bacterial CFU that were successful at invading them (output). In the case of *S. flexneri mxiD(Q323X)*, the CFU count was not above the technical limit of observation ($10^2$ CFU/mL). (D) SDS-PAGE gel showing the total secreted proteins precipitated from the supernatants of indicated bacterial cultures after induction of secretion by Congo red. BSA was used as a control for protein precipitation. (E) Invasiveness of WT, *icsA(Q59X)*, and *vacJ(Q90X)* *S. flexneri* determined by the gentamicin protection assay. (F) Schematic representation of the plaque formation assay. (G) Intracellular dissemination of WT and mutant *S. flexneri* determined by plaque formation assay. The images

To verify this effect at the mRNA level, we additionally mutated *ipgB*, a gene upstream from *ipaC*, by constructing two *ipgB* mutant strains [*S. flexneri ipgB*(Q44X) and *S. flexneri ipgB*(Q190X)]. We determined the expression of *icsB*, *ipgB*, and *ipaC* by quantitative PCR (qPCR) (Fig. 6A). Two-way analysis of variance (ANOVA) showed no significant differences between the transcription of *ipgB* and *ipaC* between wild-type *S. flexneri* and the four *icsB* and *ipgB* mutants (Fig. 6B). Moreover, no mutant showed significant differences compared to wild-type *S. flexneri* when transcription of *ipgB* or *ipaC* was analyzed individually (Fig. S4D). As expected, no effect was observed at the protein level, assessed by comparing the secretion of IpaC (Fig. 6C). Although a loss-of-function mutation of *ipgB* has been reported to be 50% less invasive (41), we did not observe this decrease, mainly because this reduction is within the experimental error in our hands (Fig. 6D). We did observe a significant effect on the levels of *icsB* transcription (Fig. S4D), particularly for the *S. flexneri icsB*(Q32X) mutant, which is consistent with our previous report (42) that some mutations, including premature stop codons, can decrease mRNA levels without affecting observed phenotypes. Altogether, these results showed that base-edited premature stop codon mutations did not result in any observable polar effects, at least at the phenotype level.

## DISCUSSION

Functional characterization of genes is a primary step in understanding the physiology and pathogenesis of microbes, which is made possible by powerful tools like high-throughput reverse genetics (43). However, the success of this approach is entirely dependent on the ability to generate loss-of-function libraries with an equally high throughput (44). By using a two-plasmid system and replacing electroporation with chemical transformation, we were able to increase throughput, ensuring pure colonies of several desired mutants prepared in parallel in just 5 to 6 days from the receipt of spacer oligonucleotides (Fig. S5). The use of the type IIs restriction enzyme BsaI not only ensured directional cloning of spacer oligonucleotides but also provided the advantage that the parent plasmid needed to be digested only once for all cloning reactions. Furthermore, even for the gRNA that showed no significant difference in the frequency of mutation (sgRNA_m3) compared to that of the control, the reaction yielded a workable number of mutant colonies.

Although high frequencies of mutation were achieved in previous studies by combining the expression of catalytically inactive Cas9, dCas9-AID, and a uracil DNA glycosylase inhibitor (21, 45), we avoided using this combination because the uracil DNA glycosylase inhibitor can potentially induce hypermutation due to its ability to impair DNA repair (46). In our case, the apparently low stability of nCas9-AID and combined expression with an sgRNA, targeting or nontargeting, might have decreased the availability of free apo-nCas9-AID, preventing it from binding to nonspecific sites on genomic DNA (47) and introducing unwanted, potentially toxic mutations. Since the virulence plasmid is not necessary for bacterial survival, improved yield of mutants by counterselecting the nonmodified cells using CRISPR-Cas9 and recombineering methods together is not applicable to mutation of plasmid-borne genes (16, 17). However, we were able to mutate plasmid-borne and chromosomal genes in *S. flexneri* in parallel using the same strategy and workflow.

It is now clear that polar effects are prevalent even in well-characterized systems like the Keio collection (48). Therefore, even if we did not observe any polar effects at the phenotype level in this study, we observed potential changes in mRNA abundance and recommend that such effects are taken into account while generating base-edited mutants, in the same manner as for any other mutagenesis strategy.

**FIG 5** Legend (Continued)

depict the stained monolayer of TC7 cells in a 6-well plate with the plaques seen as zones of clearance. (H) Invasiveness of the WT and the two strains of *S. flexneri icsB* mutants determined by gentamicin protection assay. (I) Measurement of the diameters of the plaques from one representative plaque formation assay. In all the plots, each dot represents a technical replicate and is color coded to represent a biological replicate. The bars represent the median values from three independent experiments, except in panel I, where the bars represent the median diameter values from one representative image. Statistical significance was determined by performing two-way ANOVA followed by Sidák's multiple comparison test. ns, not significant; *, $P < 0.0332$; **, $P < 0.0021$; ***, $P < 0.0002$; ****, $P < 0.0001$.

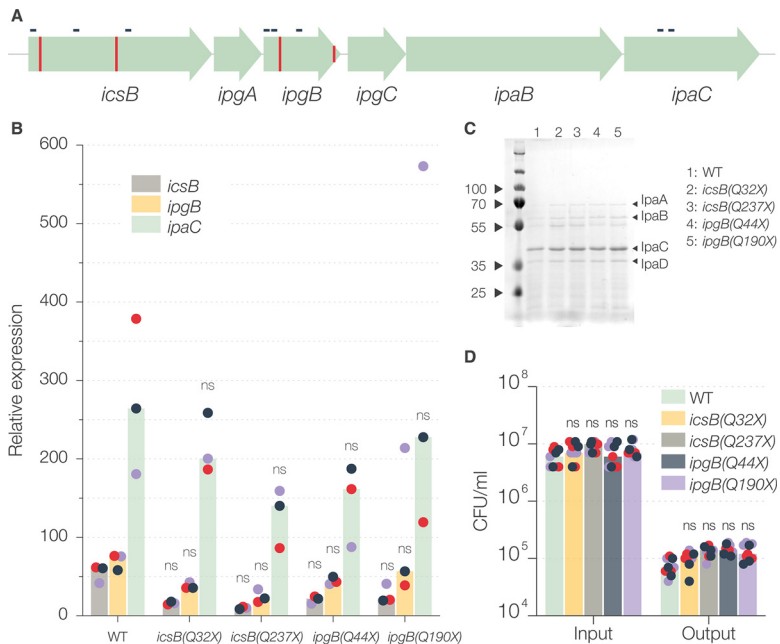

**FIG 6** Assessment of polar effects of base-edited mutations. (A) Representation of the *icsB* operon. The red vertical lines represent the sites of introduction of stop codons in *icsB* and *ipgB* genes. The small horizontal black lines represent the amplicons used to determine the mRNA abundance by qPCR. (B) Normalized (to *cysG* and *hcaT*) expression of *icsB*, *ipgB*, and *ipaC* genes in different mutant strains. Each dot represents a single amplicon and is color coded to represent a biological replicate. Statistical significance was determined by performing two-way ANOVA and Holm-Sidák's multiple comparison test compared to the WT. ns, not significant; *, $P < 0.0332$; **, $P < 0.0021$; ***, $P < 0.0002$; ****, $P < 0.0001$. (C) SDS-PAGE gel showing the secreted proteins (labeled with arrowheads) precipitated from the supernatants of indicated bacterial cultures after induction of secretion by Congo red. (D) Invasiveness of *S. flexneri* strains, determined by the number of bacterial CFU used to infect TC7 cells (input) and the number of bacterial CFU that were successfully invaded (output). Each dot represents a technical replicate and is color coded to represent a biological replicate. The bars represent the median values from three biological replicates. Statistical significance was determined by performing two-way ANOVA followed by Sidák's multiple comparison test compared to the WT. ns, not significant; *, $P < 0.0332$; **, $P < 0.0021$; ***, $P < 0.0002$; ****, $P < 0.0001$.

In addition to generating parallel single gene mutations, a spacer oligonucleotide library can be cloned into pgRNA in a single reaction, resulting in a pgRNA plasmid library. This library can be transformed into bacteria expressing nCas9-AID, generating an equally large pooled library of loss-of-function mutants. Since the pgRNA (ColE1-derived origin) propagates stably (49), cloned spacer oligonucleotides can function as barcodes, aiding in high-throughput screening of gene function (50), and base editing can subsequently be used for targeted mutagenesis of selected candidates.

Despite many apparent advantages, the most obvious limitation of the base-editing method is the availability of mutable sites in a particular gene. Certain genes, for example, *mxiD* in this study, have only one mutable site. As seen from our results, certain sgRNAs are more effective than others, and not having any alternative can therefore be limiting. However, with the discovery of Cas9 variants with various PAM specificities, there is a possibility of combining them with deaminases to expand the arsenal of effector proteins to overcome this limitation (51). Altered PAM specificity can also address another possible limitation in the form of usage of the amber codon for termination. The amber codon is only rarely used for termination of essential genes and is often a suboptimal stop codon (52).

In conclusion, we have successfully adapted a programmable CRISPR-Cas9 base-editing method to generate loss-of-function mutants in *S. flexneri* and *E. coli*. This method can allow functional characterization of unknown *S. flexneri* genes by high-throughput reverse genetics and could be extended to other members of the *Enterobacteriaceae* due to plasmid and promoter compatibility.

## MATERIALS AND METHODS

**Bacterial strains, cells, and culture conditions.** Bacterial strains, plasmids, and cells used in this study have been summarized in Table S1. All *E. coli* strains were grown in LB, *S. flexneri* in tryptic soy broth (TSB), and TC7 cells were grown in Dulbecco modified Eagle medium (DMEM) containing 10% fetal bovine serum (FBS), nonessential amino acids, and penicillin-streptomycin. For maintenance of plasmids in *E. coli*, chloramphenicol and carbenicillin were used at 25 $\mu$g/mL and 100 $\mu$g/mL, respectively. In the case of *S. flexneri*, the concentrations were 15 $\mu$g/mL and 50 $\mu$g/mL, respectively. Spectinomycin was used at 50 $\mu$g/mL. *S. flexneri* colonies were selected on agar plates supplemented with 0.01% Congo red.

**Construction of fluorescent strains.** Fluorescent strains of *E. coli* and *S. flexneri*, constitutively expressing *mCherry* (and *mScarlet-i* in the case of *Shigella*) from the chromosome, were constructed using Tn*7*-mediated transposition (53, 54). The detailed methodology is given in Text S1.

**Plasmid construction.** All the oligonucleotides used for recombination procedures are listed in Table S2. pgRNA_AT was generated by combining parts from pgRNA-ccdB and a pBluescript SK variant that lacks the BsaI site in the $\beta$-lactamase gene (22). pnCas9-AID was generated by combining parts of pSU19 and pdCas9-AID by *in vivo* assembly (IVA) cloning (22, 55). Plasmid construction is explained in detail in Text S1.

**gRNA design and cloning.** The guide RNAs (gRNAs) for mutagenesis were designed using the program CRISPR_CBEI (26) (Table S2). Details of cloning and guide RNA design are given in Text S1.

**Mutagenesis.** pnCas9-AID and the respective pgRNAs were cotransformed into ultracompetent *E. coli* cells using the heat shock method (56). The cotransformants were selected in Luria-Bertani (LB) agar plates supplemented with chloramphenicol, carbenicillin, and glucose. Since cotransformation was not very efficient in *S. flexneri*, pnCas9-AID and various sgRNA plasmids were sequentially transformed.

Bacteria carrying both plasmids were grown overnight at 37°C in broth supplemented with chloramphenicol, carbenicillin, and glucose. The overnight culture was diluted 100 times in fresh medium, and the bacteria were grown until reaching an optical density at 600 nm (OD$_{600}$) of 0.2. A concentration of 1 mM IPTG was added to overexpress nCas9-AID, and the cultures were incubated at 37°C for another 2 h (longer incubation times were tested in the cases of *mCherry* and *gyrA* mutagenesis). Tenfold serial dilutions of the cultures were made in sterile phosphate-buffered saline (PBS), and 10 $\mu$L of each dilution was spotted on agar plates containing no antibiotics. For *gyrA* mutagenesis, the agar plates were supplemented with nalidixic acid (30 $\mu$g/mL) and, in addition to 10-fold dilutions, a 10-fold-concentrated culture was also spotted. The agar plates were incubated overnight at 37°C, and the colonies were enumerated. The plates were imaged in a Typhoon scanner (GE Healthcare) to determine fluorescence. Four random colonies were selected in each case to verify mutation by PCR and sequencing.

**Fluorescence measurement and microscopy.** Overnight cultures of *E. coli* MG1655 and *S. flexneri*::*mScarlet* carrying pgRNA-(R)-GFP were diluted 100 times in fresh medium containing carbenicillin. Amounts of 100 $\mu$L of culture were pipetted into different wells of a 96-well plate, and the fluorescence (excitation, 480 nm, and emission, 510 nm) and OD$_{600}$ were measured simultaneously every 10 min in a Spark multimode plate reader (Tecan) at 37°C. After 3 h of incubation, 10-$\mu$L amounts of both cultures were mixed in a tube and spotted onto a thin agarose pad on a glass slide. Microscopy images were taken using a wide-field microscope (Nikon) with differential interference contrast (DIC), Texas red, and fluorescein isothiocyanate (FITC) filters. The images were given false colors and fluorescence quantification was performed using ImageJ.

**Immunodetection of nCas9-AID.** Bacteria were cultured similarly to the mutagenesis assay, but instead of spotting dilutions on agar plates, 10-mL amounts of cultures were pelleted by centrifugation and resuspended in 250 $\mu$L of 1$\times$ Laemmli buffer. The samples were boiled and resolved on 10% SDS-PAGE gels. The gels were stained using SimplyBlue stain (Thermo Fisher) or used for transfer of proteins onto polyvinylidene difluoride (PVDF) membranes (GE Healthcare). Probing was done using anti-Cas9 primary antibody (catalog no. PA5-90171; Thermo Fisher) at 1:20,000 dilution and horseradish peroxidase (HRP)-conjugated anti-rabbit secondary antibody (Abcam) at 1:40,000 dilution. The blot was developed using enhanced chemiluminescence (ECL) reagent (Cytiva) and imaged using the Amersham imager 680 (Sigma-Aldrich).

**Bacterial invasion assay and plaque formation assay.** The ability of bacteria to invade epithelial cells was determined by gentamicin protection assay as described earlier (33). All mutant *Shigella* strains were transformed with pAfaE, a derivative of pIL22 (57) expressing AFA I adhesin. The plaque formation assay was performed as described previously (38), with modifications as follows. TC7 cells were infected at an MOI of 1:500 (bacteria/cells) and additionally at 1:150 for *vacJ* and *icsA* mutants. The infected cells were incubated for 48 to 72 h before fixation and staining.

**Secretion assay.** Secretion assays were performed as described earlier, with modifications (58). Overnight cultures of bacteria were diluted 200 times and were grown until reaching an OD$_{600}$ of 0.5. Cells were collected from 4 mL culture by centrifugation and washed twice with PBS. The pellet was finally resuspended in 1 mL prewarmed PBS containing 0.01% Congo red and incubated for 1 h at 37°C. Bacteria were separated by centrifugation, and the proteins in the supernatants were precipitated at 4°C for 10 min using 20% trichloroacetic acid. A 1-mg/mL solution of bovine serum albumin (BSA) was used as a control for precipitation. The precipitated proteins were washed twice with ice-cold acetone and resuspended in Laemmli buffer before being resolved on a 12% polyacrylamide gel. The gel was soaked overnight in 50% ethanol to remove Congo red and rehydrated gradually. The gel was stained with SimplyBlue SafeStain (Thermo Fisher) to visualize the proteins.

**96-Well mutagenesis.** Amounts of 50 $\mu$L of chemically competent *S. flexneri* carrying pnCas9-AID were pipetted into wells of a prechilled 96-well plate. sgRNA plasmids were added to individual wells and transformed by the heat shock method. An amount of 150 $\mu$L of super optimal broth with catabolite repression (SOC) was added to each well, and the plate was incubated at 37°C for 1 h. Amounts of 100 $\mu$L of cell suspensions and 1/10th dilution from each well were spread onto selective plates. The plates were incubated overnight at 37°C, and single colonies were picked up for mutagenesis. In a 96-well deep-well plate (Eppendorf), single colonies were inoculated into 500 $\mu$L of TSB with antibiotics and grown overnight at 37°C. Amounts of 10 $\mu$L of overnight culture were added to 500 $\mu$L of fresh medium in a new 96-well deep-well plate, which was incubated until reaching an $OD_{600}$ of 0.2. A concentration of 1 mM IPTG was added to the wells, and the plate was incubated for another 2 h before serial dilutions (10 $\mu$L of $10^{-5}$ and $10^{-6}$ dilutions) of the cultures were spotted on Congo red plates. The plates were incubated at 37°C overnight, and single colonies were checked for the desired mutations by colony PCR and DNA sequencing.

**qPCR analysis.** Total RNA extraction and qPCR were performed as previously described, with minor modifications (42, 59). Bacteria were grown similarly to the mutagenesis experiment, but instead of spotting the dilutions, the cultures were treated with ice-cold stop solution (95% [vol/vol] ethanol, 5% [vol/vol] phenol, pH 4) for 30 min on ice. The cells were then collected by centrifugation and stored at $-80$°C. cDNAs were subsequently prepared with random hexamer primers using a first-strand cDNA synthesis kit (Thermo Fisher). To analyze the expression of *icsB*, *ipgB*, *ipaC*, and two housekeeping genes, *hcaT* and *cysG* (60), we designed specific oligonucleotides (Table S2). To analyze qPCR data, we used the iQ5 optical system software (Bio-Rad) and obtained the cycle threshold ($C_T$) value with default settings. We manually verified that changes in the threshold did not affect or bias the results. To normalize the data, the average $C_T$ values of the two housekeeping genes were used to calculate a $\Delta C_T$ value for the gene of interest. Controls with no reverse transcriptase were also analyzed to exclude the presence of contaminating genomic DNA fragments.

## SUPPLEMENTAL MATERIAL

Supplemental material is available online only.

**TEXT S1**, DOCX file, 0.03 MB.
**FIG S1**, TIF file, 7.4 MB.
**FIG S2**, TIF file, 8.7 MB.
**FIG S3**, TIF file, 4.6 MB.
**FIG S4**, TIF file, 8.0 MB.
**FIG S5**, TIF file, 1.8 MB.
**TABLE S1**, DOCX file, 0.04 MB.
**TABLE S2**, DOCX file, 0.03 MB.

## ACKNOWLEDGMENTS

A.P. acknowledges funding from Swedish Research Council (Vetenskapsrådet) grant number #2016-06598, institutional support from Umeå University, Knut and Alice Wallenberg Foundation grant number KAW 2015.0225, and Carl Kempe Foundation grant number JCK-2031.3. D.A.C. acknowledges support from Carl Tryggers Stiftelse för Vetenskaplig Forskning grant number CTS 18-65 and Kempestiftelserna grant number SMK 1860. D.A.C. was supported in part also by funds from the Novo Nordisk Foundation (grant number NNF17OC0026486) awarded to Emmanuelle Charpentier at MIMS, The Laboratory for Molecular Infection Medicine Sweden. A.S. was funded by a MIMS Excellence by Choice postdoctoral program stipend under the patronage of Emmanuelle Charpentier, grant number SMK-1532.2, and Svenska Sällskapet för Medicinsk Forskning (SSMF) postdoctoral grant number PD20-0022.

We thank Yu Wang and Jibin Sun, Tianjin Institute of Industrial Biotechnology, Chinese Academy of Sciences, Tianjin, China, for kindly gifting the pnCas9-AID-YU and pgRNA-ccdB plasmids, Javier Pizarro-Cerdá, Institut Pasteur, Paris, France, for providing the pSU2.1rp-mCherry plasmid, and Justin L. Sonnenburg, Stanford University School of Medicine, Stanford, CA, USA, for providing the sfGFP-carrying *Bacteroides thetaiotaomicron* strain.

Author contributions according to CASRAI are as follows. Conceptualization: A.S., A.P., D.A.C. Formal analysis: A.S., R.O.A., D.A.C. Funding acquisition: A.S., A.P., D.A.C. Investigation: A.S., R.O.A. Methodology: A.S., R.O.A., A.P., D.A.C. Project administration: A.P., D.A.C. Resources: A.P., D.A.C. Supervision: A.P., D.A.C. Visualization: A.S., R.O.A. Writing – original draft: A.S. Writing – reviewing and editing: A.P., D.A.C., A.S.

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
