## [Reviewer comments · mSystems]

CRISPR-Cas guided mutagenesis of chromosome and virulence plasmid in *Shigella flexneri* by cytosine base editing

Atin Sharma, Ruqiya Omer Aden, Andrea Puhar, and David Cisneros

Corresponding Author(s): David Cisneros, Umeå University

Review Timeline:

Submission Date:

October 24, 2022

Accepted:

November 22, 2022

Editor: Mariana Byndloss

Reviewer(s): Disclosure of reviewer identity is with reference to reviewer comments included in decision letter(s). The following individuals involved in review of your submission have agreed to reveal their identity: François-Xavier Campbell-Valois (Reviewer #1)

Transaction Report:

DOI: <https://doi.org/10.1128/msystems.01045-22>

November 22, 2022

Dr. David A. Cisneros
Umeå University
Molecular Biology
Building 6L-K
Umeå, Västerbotten 90187
Sweden

Re: mSystems01045-22 (CRISPR-Cas guided mutagenesis of chromosome and virulence plasmid in *Shigella flexneri* by cytosine base editing)

Dear Dr. David A. Cisneros:

Your manuscript has been accepted, and I am forwarding it to the ASM Journals Department for publication. For your reference, ASM Journals' address is given below. Before it can be scheduled for publication, your manuscript will be checked by the mSystems production staff to make sure that all elements meet the technical requirements for publication. They will contact you if anything needs to be revised before copyediting and production can begin. Otherwise, you will be notified when your proofs are ready to be viewed.

Publication Fees:

If you would like to submit a potential Featured Image, please email a file and a short legend to msystems@asmusa.org. Please note that we can only consider images that (i) the authors created or own and (ii) have not been previously published. By submitting, you agree that the image can be used under the same terms as the published article. File requirements: square dimensions (4" x 4"), 300 dpi resolution, RGB colorspace, TIF file format.

We recognize that the video files can become quite large, and so to avoid quality loss ASM suggests sending the video file via <https://www.wetransfer.com/>. When you have a final version of the video and the still ready to share, please send it to mSystems staff at msystems@asmusa.org.
